# BOOTSTRAP MOTION FORECASTING WITH SELF-CONSISTENT CONSTRAINTS

## ABSTRACT

We present a novel framework to bootstrap **M**otion forecast**I**ng with **S**elf-consistent **C**onstraints (MISC). The motion forecasting task aims at predicting future trajectories of vehicles by incorporating spatial and temporal information from the past. A key design of MISC is the proposed Dual Consistency Constraints that regularize the predicted trajectories under spatial and temporal perturbation during training. Also, to model the multi-modality in motion forecasting, we design a novel self-ensembling scheme to obtain accurate teacher targets to enforce the self-constraints with multi-modality supervision. With explicit constraints from multiple teacher targets, we observe a clear improvement in the prediction performance. Extensive experiments on the Argoverse motion forecasting benchmark show that MISC significantly outperforms the state-of-the-art methods. As the proposed strategies are general and can be easily incorporated into other motion forecasting approaches, we also demonstrate that our proposed scheme consistently improves the prediction performance of several existing methods.

## 1 INTRODUCTION

Motion forecasting has been a crucial task for self-driving vehicles that aims at predicting the future trajectories of agents (e.g., cars, pedestrians) involved in the traffic. The predicted trajectories can further help self-driving vehicles to plan their future actions and avoid potential accidents. Since the future is not deterministic, motion forecasting is intrinsically a multi-modal problem with substantial uncertainties. This implies that an ideal motion forecasting method should produce a distribution of future trajectories or at least multiple most likely ones.

Due to the inherent uncertainty, motion forecasting remains challenging and unsolved yet. Recently, researchers have proposed different architectures based on various representations to encode the kinematic states and context information from HDMap in order to generate feasible multi-modal trajectories (Bansal et al., 2019; Chai et al., 2019; Gao et al., 2020; Gu et al., 2021; Liang et al., 2020; Liu et al., 2021; Ngiam et al., 2021; Varadarajan et al., 2021; Ye et al., 2021; Zeng et al., 2021; Zhao et al., 2020). These methods follow a traditional static training pipeline, where frames of each scenario are split into historical frames (input) and future frames (ground truth) in a fixed pattern. Nevertheless, the prediction task is a streaming task in real-world applications, where the current state will become a historical state as time goes by, and the buffer of the historical state is a queue structure to make successive predicted trajectories. As a result, the temporal consistency thus becomes a crucial requirement for the downstream tasks for fault and noise tolerance. To tackle this issue, trajectory stitching is widely applied in traditional planning algorithms (Fan et al., 2018) to ensure stability along the temporal horizon. However, as the trajectory stitching operation is non-differentiable, it cannot be easily incorporated into learning-based models. Though deep-learning-based models show unprecedented motion prediction performance compared with traditional counterparts, they do not explicitly consider the temporal consistency, leading to unstable behaviors in downstream tasks such as planning.

Inspired by these phenomena, we raise a question: can we explicitly enforce the consistency when training a deep motion prediction model? On the one hand, the predicted trajectories should be consistent given the successive inputs along the temporal horizon, namely temporal consistency. On the other hand, the predicted trajectories should be stable and robust against small spatial noise

or disturbance, namely spatial consistency. In this work, we propose a self-supervised scheme to enforce consistency constraints in both spatial and temporal domains, namely Dual Consistency Constraints. Our proposed framework, referred as ***MISC***, significantly improves the quality and robustness of motion forecasting, without the need for extra data.

On top of the consistency, multi-modality is another core characteristic of the motion prediction task. Existing datasets (Chang et al., 2019; Sun et al., 2020) only provide a single ground-truth trajectory for each scenario, which can not satisfy the multi-choice situations such as junction scenarios. Most methods adopt the winner-takes-all (WTA) (Lee et al., 2016) or its variants (Breuer et al., 2021; Narayanan et al., 2021) to alleviate this situation. However, WTA tends to produce confused predictions when two trajectories are very close. In contrast, our method addresses the multi-modality issue by introducing more powerful teacher targets from self-ensembling. With self-constraint from multiple soft teacher targets, our model is more likely to be exposed to more high-quality samples, bootstrapping each modality.

Our contributions are summarized as follows,

- We propose Dual Consistency Constraints to enforce temporal and spatial consistency in our model, which is shown to be a general and effective way to improve the overall performance in motion forecasting.
- We propose a self-ensembling constraints training strategy that provides multi-modality supervision explicitly during training to enforce self-consistency with teacher targets.
- We conduct extensive experiments on the Argoverse (Chang et al., 2019) motion forecasting benchmark and our proposed approach achieves the state-of-the-art results.

## 2 RELATED WORK

**Motion Forecasting.** Traditional methods (Houenou et al., 2013; Schulz et al., 2018; Xie et al., 2017; Ziegler et al., 2014) for motion forecasting mainly utilize HDMap information for the prior estimation and Kalman filter (Kalman, 1960) for motion states prediction. With the recent progress of deep learning on big data, more and more works have been proposed to exploit the potential of data mining in motion forecasting. Methods (Bansal et al., 2019; Chai et al., 2019; Duvenaud et al., 2015; Gao et al., 2020; Henaff et al., 2015; Liang et al., 2020; Liu et al., 2021; Shuman et al., 2013; Song et al., 2021; Ye et al., 2021; Zeng et al., 2021) explore different representations, including rasterized image, graph representation, point cloud representation and transformer to generate the features for the task and predict the final output trajectories by regression or post-processing sampling. Most of these works focus on finding more effective and compact ways of feature extraction on the surrounding environment (HDMap information) and agent interactions. Based on these representations, other approaches (Casas et al., 2018; Mangalam et al., 2020; Song et al., 2021; Zeng et al., 2021; 2019; Zhao et al., 2020) try to incorporate the prior knowledge with traditional methods, which take the predefined candidate trajectories from sampling or clustering strategies as anchor trajectories. To some extent, these candidate trajectories can provide better guidance and goal coverage for the trajectories regression due to straightforward HDMap encoding. Nevertheless, this extra dependency makes the stability of models highly related to the quality of the trajectory proposals. Goal-guided approaches (Gilles et al., 2021; Gu et al., 2021; Gilles et al., 2022) are therefore introduced to optimize goals in an end-to-end manner, paired with sampling strategies that generate the final trajectory for better coverage rate.

**Consistency Regularization.** Consistency Regularization has been fully studied in semi-supervised and self-supervised learning. Temporally related works (Wang et al., 2019; Lei et al., 2020; Zhou et al., 2017) have widely explored the idea of cyclic consistency. Most of the works apply pairwise matching to minimize the alignment difference through optical flow or correspondence matching to achieve temporal smoothness. Other works (Bachman et al., 2014; Földiák, 1991; Ouyang et al., 2021; Sajjadi et al., 2016; Wang et al., 2021) apply consistency constraints to predictions from the same input with different transformations in order to obtain perturbation-invariant representations. Our work can be seen as a combination of both types of consistency to fully consider the spatial and temporal continuity in motion forecasting.

**Multi-hypothesis Learning.** Motion forecasting task inherently has multi-modality due to the future uncertainties and difficulties in acquiring accurate ground-truth labels. WTA (Guzman-Rivera

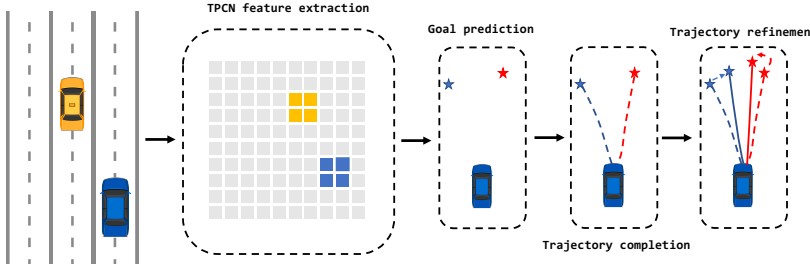

Figure 1: The overall architecture. We utilize TPCN as a feature extraction backbone to model the spatial and temporal relationship among agents and map information. A goal prediction header is then used to regress the possible goal candidates; with the goal position, we apply trajectory completion to obtain full trajectories; finally, the trajectories are refined based on the output of the trajectory completion module as anchor trajectories.

et al., 2012; Sriram et al., 2019) in multi-choice learning and its variants (Makansi et al., 2019; Rupprecht et al., 2017) incorporate with better distribution estimation to improve the training convergence, thus allowing more multi-modality. Some anchor-based methods (Breuer et al., 2021; Chai et al., 2019; Phan-Minh et al., 2020; Zeng et al., 2021) introduce pre-defined anchors based on kinematics or road graph topology to provide guidance. However, these methods only allow one target per training stage. Other methods (Breuer et al., 2021; Gu et al., 2021) try to generate multi-target for supervision with heavy handcrafted optimizations. We propose a Teacher-Target-Constraints approach to provide more precise trajectory teacher labels by leveraging the power of self-ensembling (Lee et al., 2013; Zheng et al., 2021). Multiple targets are explicitly provided to each agent to better model the multi-modality.

## 3 APPROACH

The overall architecture of MISC consists of three parts. 1) We first utilize a joint spatial and temporal learning framework TPCN (Ye et al., 2021) to extract pointwise features. Based on these features, we decouple the trajectory prediction problem as a two-stage regression task. The first stage performs goal prediction and completes the trajectory with the goal position guidance. The second stage takes the output of the first stage as anchor trajectories for refinement. 2) To train our MISC, we propose **Dual Consistency Constraints** to regularize the predictions both spatially and temporally in a streaming task view. 3) We generate more accurate teacher targets by self-ensembling to provide self-consistent **Teacher Targets Constraints** in Sec. 3.3.

### 3.1 ARCHITECTURE

Recently, TPCN (Ye et al., 2021) has gained popularity in this task due to its flexibility for joint spatial-temporal learning and scalability to adopt more techniques from point cloud learning. Considering its inferiority in representing future uncertainty, we extend TPCN with a two-stage manner through goal position prediction for more accurate waypoints prediction as our baseline. The whole network is shown in Fig. 1.

**Feature Extraction:** TPCN utilizes dual-representation point cloud learning techniques with multi-interval temporal learning to model the spatial and temporal relationship. All the historical trajectories of input agents and map information are based on pointwise representation $\{\mathbf{p}_1, \mathbf{p}_2, \ldots, \mathbf{p}_N\}$, where $\mathbf{p}_i$ is the $i$-th point with $N$ points in total, and then go through multi-representation learning framework to generate pointwise features $\mathcal{P} \in R^{N \times C}$, where $C$ is the channel number.

**Goal Prediction:** With the pointwise features from the backbone, we also adopt the popular goal-based ideas (Gilles et al., 2021; Gu et al., 2021; Zhao et al., 2020) to find the optimal planning policy. Specifically, we first gather all corresponding pointwise agent features and then sum over features to get the agent instance feature $\phi \in R^{1 \times C}$. To generate $K$ goal position prediction $G = \{G^k : (g_x^k, g_y^k) | 1 \leq k \leq K\}$, we use a simple MLP layer: $G = MLP(\phi)$. Instead of relying

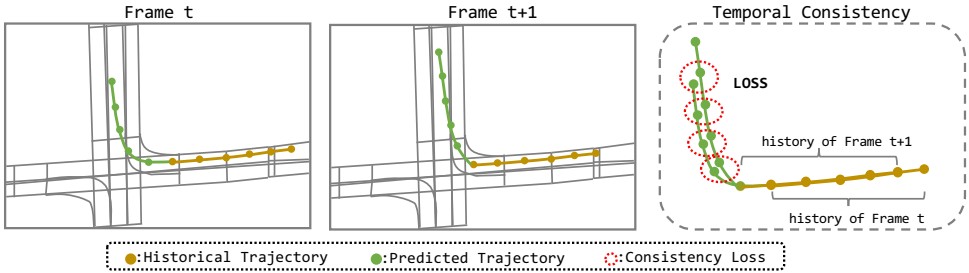

Figure 2: The overall idea of the temporal consistency. In the training stage, we first generate output prediction trajectory points as normal for each given scenario. Then we slide the input with a step in order to introduce the streaming nature to generate the consecutive output trajectory points. The proposed temporal consistency requires the overlap between these two outputs to be consistent

on heavy sampling strategies like previous goal-based methods, our method avoids generating extra proposals, which may lead to a large computation overhead.

**Trajectory Completion:** With the predicted goal positions, we need to complete each trajectory conditioned on these goals. We propose a simple trajectory completion module to generate $K$ full trajectories $\left\{\tau_{reg}^k | 1 \leq k \leq K\right\}$ with a single MLP layer as follows:

$$\tau_{reg}^k = \{(x_1^k, y_1^k), (x_2^k, y_2^k), \ldots, (x_T^k, y_T^k)\} = MLP(concat(\phi, G^k)). \tag{1}$$

**Trajectory Refinement:** Inspired by Faster-RCNN (Ren et al., 2015) and Cascade-RCNN (Cai & Vasconcelos, 2018), we use the output trajectories from the Trajectory Completion as anchor trajectories to refine trajectories and predict the corresponding possibility of each trajectory. In particular, the input of the trajectory refinement module will be the whole trajectory with agent historical waypoints $\tau_{history}$. With a residual block followed by a linear layer $Reg$ and $Cls$ respectively, we regress the delta offset to the first stage outputs $\Delta_{\tau_{reg}} = Reg(\tau_{reg}, \tau_{history})$ and corresponding scores $\tau_{cls} = \left\{c^k | 1 \leq k \leq K\right\}$ respectively, where $\tau_{cls} = Cls(\tau_{reg}, \tau_{history})$. The final output trajectories will be $\tau_{reg'} = \Delta_{\tau_{reg}} + \tau_{reg}$.

### 3.2 DUAL CONSISTENCY CONSTRAINTS

Consistency regularization has been proved as an effective self-constraint that helps improve robustness against disturbances. Therefore, we propose **Dual Consistency Constraints** in both spatial and temporal domains to align predicted trajectories for continuity and stability.

#### TEMPORAL CONSISTENCY

In motion forecasting, since each training scenario contains multiple successive frames within a fixed temporal chunk, it is reasonable to assume that any two overlapping chunks of input data with a small time-shift should produce consistent results. The motion forecasting task aims to predict $K$ possible trajectories with $T$ time steps for one scenario, given $M$ frames historical information. Suppose the information at each history frame is $I_i$, where $1 \leq i \leq M$ and the $k$-th output future trajectories are $\left\{(x_i^k, y_i^k) | M < i \leq M + T\right\}$. We first apply time step shift $s$ for the input for temporal consistency. Therefore, the input history frames information will be $\{I_i | 1 + s \leq i \leq M + s\}$ and then we apply the same network for the shifted history information with surrounding HDMap information to generate the $k$-th output trajectories $\left\{(x_i'^k, y_i'^k) | M + s < i \leq M + s + T\right\}$. When $s$ is small, the driving intentions or behavior keeps stable in a short period. Since both trajectories have $T - s$ overlapping waypoints, they should be as close as possible and share consensus. Thus, we can construct self-constraints for a single scenario input due to the streaming property of the input data. Fig. 2 demonstrates the overall idea of the temporal consistency constraint.

**Trajectory Matching:** Since we predict $K$ future trajectories to deal with the multi-modality, it is crucial to consider the trajectory matching relationship between original predictions and time-shifted predictions when applying the temporal consistency alignment. For a matching problem, the metric on similarity criteria and matching strategies will be two key factors. Several ways can be used to

measure the difference between trajectories, such as Average Displacement Error (**ADE**) and Final Displacement Error (**FDE**). We utilize **FDE** as the criteria since the last position error can partially reflect the similarity with less bias from averaging compared with **ADE**.

**Matching Strategy:** There are roughly four ways used for matching, namely forward matching, backward matching, bidirectional matching, and Hungarian matching. Forward matching takes one trajectory in the current frame and finds its corresponding trajectory in the next frame with the least cost or maximum similarity. Backward matching is the reverse way compared to forward matching. Furtherly, bidirectional matching consists of both forward and backward matching, which considers the dual relationship. Hungarian matching is a linear optimal matching solution based on linear assignment. Forward and backward matching only considers the one-way situation, which is sensitive to noise and unstable. Hungarian matching has a high requirement for cost function choice. Based on these observations, we choose bidirectional matching as our strategy. We also show its advantages over the other approaches in Sec. 4.3.

After obtaining the optimal matching pairs $\{(m_k, n_k) | 1 \leq k \leq K\}$, we can compute the consistency constraint by a simple smooth $L_1$ loss (Ren et al., 2015) $\mathcal{L}_{Huber}$:

$$\mathcal{L}_{temp} = \sum_{k=1}^{K} \sum_{t=s+1}^{T} \mathcal{L}_{Huber}((x_t^{m_k}, y_t^{m_k}), (x'^{n_k}_{t-s}, y'^{n_k}_{t-s})). \tag{2}$$

SPATIAL CONSISTENCY

Since our MISC is a two-stage framework, the second stage mainly aims for trajectory refinement. It will be more convenient to add spatial permutation in the second stage with less computational cost. First, we apply spatial permutation function $Z$, including flipping and random noise, to the trajectories from the first stage. The refinement module will process these augmented inputs and generate the offset to the ground truth and classification scores. Under the small spatial permutation and disturbance, we assume that the outputs of the network should also be self-consistent, meaning that the outputs have strong stability or tolerance to noise. Compared with data augmentation, it is the explicit regularization. Then the spatial consistency constraint $\mathcal{L}_{spa}$ is as follows:

$$\mathcal{L}_{spa} = \mathcal{L}_{Huber}(\Delta_{\tau_{reg}}, Z^{-1}(Reg(Z(\tau_{reg}, \tau_{history})))). \tag{3}$$

Then the total loss for Dual Consistency Constraints module will be $\mathcal{L}_{cons} = \mathcal{L}_{spa} + \mathcal{L}_{temp}$.

## 3.3 TEACHER-TARGET CONSTRAINTS

Existing datasets (Chang et al., 2019; Sun et al., 2020) only provide a single ground-truth trajectory for the target agent, which is to be predicted in one scenario. In order to encourage the multi-modality of models, the winner-takes-all (WTA) strategy is commonly used to prevent the model from collapsing into a single domain. However, the WTA training strategy suffers from instability associated with network initialization. Some other approaches (Breuer et al., 2021; Narayanan et al., 2021) introduce robust estimation methods to select better hypotheses. To some extent, these methods can only implicitly model the multi-modality. Some other approaches (Breuer et al., 2021; Zhao et al., 2020) generate several possible future trajectories based on the kinematics model and road graph topology. DenseTNT (Gu et al., 2021) only uses teacher labels for goal set prediction through a hill-climbing algorithm. These optimization methods tend to impose strict constraints and handcrafted prior knowledge, resulting in inaccurate teacher-targets and inferior performance. In contrast, our approach aims to generate more accurate teacher targets to provide explicit multi-modality supervision through self-ensembling to leverage the power of semi-supervised learning.

**Teacher-Target Generation.** The key part of our approach lies in generating more accurate teacher labels for each agent. However, it is straightforward to apply model ensembling techniques (He et al., 2020; Laine & Aila, 2016; Tarvainen & Valpola, 2017) to obtain more powerful predictions. Compared with previous works (Breuer et al., 2021; Chai et al., 2019; Zhao et al., 2020), we do not rely on handcrafted anchor trajectory sampling, which is based on inaccurate prior knowledge, including motion estimation. Meanwhile, soft targets from ensembling can better finetune the predictions and reduce the gradient variance for better training convergence. As suggested in works (Dietterich, 2000; Opitz & Maclin, 1999), the prediction error decreases when the ensemble approach is used

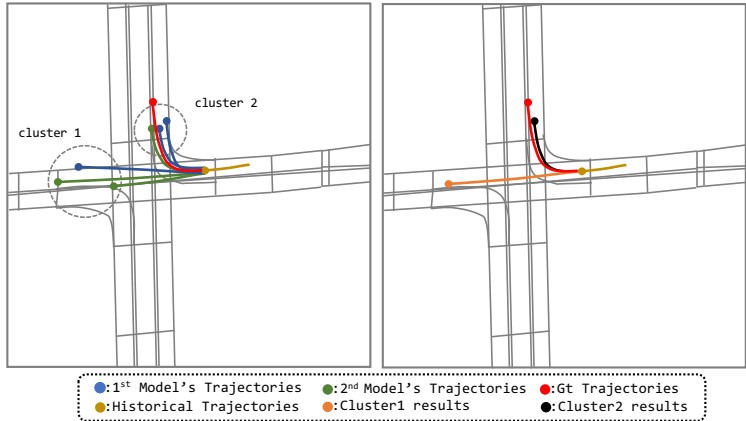

Figure 3: The overall procedure for the teacher-target generation. We obtain multiple predictions from outputs of different models for the target agents in each scenario; then we apply K-means clustering algorithm to ensemble the trajectories

once the model is diverse enough. Therefore, we apply k-means algorithm (MacQueen et al., 1967) to the predicted trajectories that are collected within different training procedures (for example, launched with different seeds of random number generators, optimized with different learning rates, etc.) of MISC without Teacher-Target Constraints to generate $J$ trajectories with corresponding scores for each scenario. Fig. 3 shows the overall process of our approach. Then with the original ground-truth label, we will formulate $J + 1$ target trajectories as follows:

$$\tau_{conf} = \{c_0, c_1, \ldots, c_J\}, \tag{4}$$

$$\tau_{tgt}^j = \{(x_1^{tgt_j}, y_1^{tgt_j}), (x_2^{tgt_j}, y_2^{tgt_j}), \ldots, (x_T^{tgt_j}, y_T^{tgt_j})\}, \tag{5}$$

where $\tau_{tgt}^j$ is the $j$-th trajectory with score $c^j$, among $J + 1$ target trajectories. To simplify the notation, $\tau_{tgt}^0$ is the ground-truth trajectory with $c_0$ set to 1.

## 3.4 LEARNING

The total supervision of our MISC can be decoupled into several parts, as described in previous sections. For the regression and classification parts, we loop over all the possible $J + 1$ targets $\tau_{tgt}$. For each target $\tau_{tgt}^j$ with confidence $\tau_{conf}^j$, we apply WTA strategy as described in Sec. 3.3. Suppose $k^*$-th trajectory from trajectory refinement output $\tau_{reg'}$ is the best trajectory which has the maximum similarity with target $\tau_{tgt}^j$, the classification loss and regression loss are defined as

$$\mathcal{L}_{cls}^j = \frac{1}{K} \sum_{k=1}^{K} \tau_{conf}^j \mathcal{L}_{Huber}(c^k, c^{k^*}), \tag{6}$$

$$\mathcal{L}_{reg}^j = \frac{1}{T} \sum_{t=1}^{T} \tau_{conf}^j \mathcal{L}_{Huber}((x_t^{k^*}, y_t^{k^*}), (x_t^{tgt_j}, y_t^{tgt_j})). \tag{7}$$

For classification loss design, we adopt the displacement prediction idea from TPCN (Ye et al., 2021) to alleviate the hard assignment phenomenon. As for converting the displacement into probability, we use the standard softmin function to distribute the scores. Since we have trajectory completion and refinement modules, the regression loss will be $\mathcal{L}_{reg} = \sum_{j=0}^{J} (\mathcal{L}_{reg}^j + \mathcal{L}_{\Delta reg}^j)$, where $\mathcal{L}_{\Delta reg}^j$ is the regression loss for the refinement module. The final loss is $\mathcal{L} = \mathcal{L}_{reg} + \mathcal{L}_{cls} + \mathcal{L}_{cons}$.

## 4 EXPERIMENTS

We conduct experiments on the Argoverse dataset (Chang et al., 2019), one of the largest publicly available motion forecasting datasets. We compare our MISC with other state-of-the-art methods. Furthermore, we provide ablation studies to evaluate the effectiveness and generalization ability of each proposed module and design experiments for some hyperparameter choices.

Table 1: The detailed results of our MISC and other top-performing approaches on the Argoverse test set. And b-FDE$_6$ is the abbreviation of brier-minFDE$_6$

| Models | minADE$_1$ | minFDE$_1$ | MR$_1$ | minADE$_6$ | minFDE$_6$ | MR$_6$ | b-FDE$_6$ |
|---|---|---|---|---|---|---|---|
| Jean (Chang et al., 2019; Mercat et al., 2020) | 1.74 | 4.24 | 0.68 | 0.98 | 1.42 | 0.13 | 2.12 |
| LaneConv (Liang et al., 2020) | 1.71 | 3.78 | 0.59 | 0.87 | 1.36 | 0.16 | 2.05 |
| LaneRCNN (Zeng et al., 2021) | 1.68 | 3.69 | 0.57 | 0.90 | 1.45 | 0.12 | 2.15 |
| mmTransformer (Liu et al., 2021) | 1.77 | 4.00 | 0.62 | 0.87 | 1.34 | 0.15 | 2.03 |
| SceneTransformer (Ngiam et al., 2021) | 1.81 | 4.06 | 0.59 | 0.80 | 1.23 | 0.126 | 1.88 |
| TNT (Zhao et al., 2020) | 1.77 | 3.91 | 0.59 | 0.94 | 1.54 | 0.13 | 2.14 |
| DenseTNT (Gu et al., 2021) | 1.68 | 3.63 | 0.58 | 0.88 | 1.28 | 0.125 | 1.97 |
| PRIME (Song et al., 2021) | 1.91 | 3.82 | 0.59 | 1.22 | 1.55 | 0.12 | 2.09 |
| TPCN (Ye et al., 2021) | 1.58 | 3.49 | 0.56 | 0.88 | 1.24 | 0.13 | 1.92 |
| HOME (Gilles et al., 2021) | 1.70 | 3.68 | 0.57 | 0.89 | 1.29 | **0.08** | 1.86 |
| MultiPath++ (Varadarajan et al., 2021) | 1.623 | 3.614 | 0.564 | 0.790 | 1.214 | 0.13 | 1.793 |
| Ours | **1.476** | **3.251** | **0.532** | **0.766** | **1.135** | 0.11 | **1.756** |

## 4.1 EXPERIMENTAL SETUP

**Dataset.** Argoverse (Chang et al., 2019) is currently one of the most popular motion forecasting datasets. It provides more than 300K scenarios with rich HDMap information. For each scenario, objects are divided into three types: agent, AV and others, where "agent" is the object to be predicted. Moreover, each scenario contains 50 frames sampled at 10 Hz, meaning that the time interval between successive frames is 0.1s. The whole dataset is split into training, validation, and test sets, with 205942, 39472, and 78143 sequences, respectively.

**Metrics.** We use the standard evaluation metrics, including ADE and FDE. ADE is defined as the average displacement error between ground-truth trajectories and predicted trajectories over all time steps. FDE is defined as displacement error between ground-truth trajectories and predicted trajectories at the last time step. We predict $K$ candidate trajectories for each scenario and calculate the metrics with the ground truth labels. Accordingly, minADE and minFDE are minimum ADE and FDE over the top $K$ predictions. Moreover, miss rate (MR) is also considered, defined as the percentage of the best-predicted trajectories whose FDE is within a threshold (2m). Brier-minFDE is the minFDE plus $(1-p)^2$, where $p$ is the corresponding trajectory probability. Metrics for $K = 1$ and $K = 6$ are used in our experiments. Note that Brier-minFDE$_6$ is the ranking metric.

**Experimental Details.** We apply some data augmentation, including random flipping with a probability 0.5 and global random scaling with the scaling ratio between $[0.8, 1.25]$ during the training stage. As for model settings, the time shift $s$ for the temporal consistency constraint is set to 1. We adopt $K = 6$ to generate 6 trajectories and use $J = 6$ teacher targets for each scenario. Furthermore, we choose bidirectional-matching for temporal consistency constraint. We finally use 10 models for ensembling due to computation resource limits. For more training details, we have included them in the supplementary materials.

## 4.2 EXPERIMENTAL RESULTS

**Argoverse Leaderboard Results.** We provide detailed quantitative results of our MISC on the Argoverse test set as well as public state-of-the-art methods in Tab. 1. Compared with previous methods, our MISC improves all the evaluation metrics except MR$_6$ by a large margin. Furtherly, since the proposed modules are all general training components, other existing motion forecasting models can also benefit greatly from these strategies.

**Qualitative Results.** We also present some qualitative results on the Argoverse validation set in Fig. 4. Compared with results without consistency, the Dual Consistency Constraints improve both the quality and smoothness of the predicted trajectories significantly, resulting in more feasible and stable results despite the input noise.

## 4.3 ABLATION STUDIES

**Component Study.** As shown in Tab. 2, we conduct an ablation study for our MISC on the Argoverse validation set to evaluate the effectiveness of each proposed component. We adopt TPCN (Ye

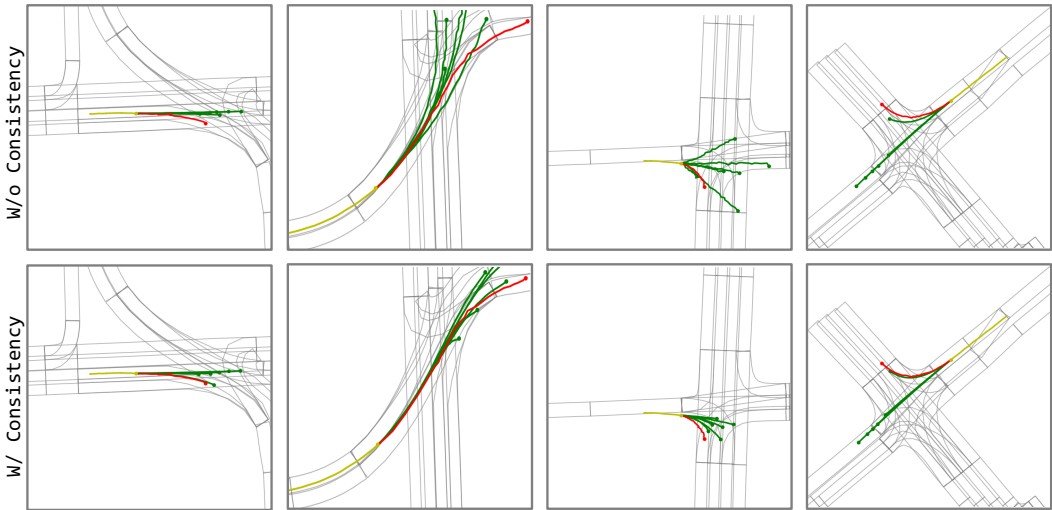

Figure 4: The past trajectory is in yellow, the predicted trajectory in green, and the ground truth in red. The top row of the figure shows the results without consistency, while the bottom row shows the results with consistency

Table 2: Ablation study results of modules. Goal refers to Trajectory completion with goal prediction. "Ref." is the trajectory refinement module, and the "Temp." is temporal consistency. TTC refers to Teacher-Target Constraints during training

| Architecture | | Consistency | | TTC | K=1 | | K=6 | |
|---|---|---|---|---|---|---|---|---|
| Goal | Ref. | Temp. | Spatial | | minADE | minFDE | minADE | minFDE |
| | | | | | 1.34 | 2.95 | 0.73 | 1.15 |
| ✓ | | | | | 1.33 | 2.91 | 0.725 | 1.10 |
| ✓ | ✓ | | | | 1.31 | 2.89 | 0.71 | 1.07 |
| ✓ | ✓ | ✓ | | | 1.24 | 2.70 | 0.662 | 0.981 |
| ✓ | ✓ | ✓ | ✓ | | 1.22 | 2.67 | 0.653 | 0.954 |
| ✓ | ✓ | | | ✓ | 1.26 | 2.77 | 0.69 | 1.01 |
| ✓ | ✓ | ✓ | ✓ | ✓ | **1.19** | **2.60** | **0.640** | **0.929** |

et al., 2021) as the baseline shown in the first row of Tab. 2 and add the proposed components progressively. The architecture modifications from the goal set prediction and trajectory refinement module show their promising improvements of about 2%. Dual consistency Constraints have the largest improvements of more than 5% among all the evaluation metrics. Especially for minFDE$_1$, temporal consistency can optimize 20 cm, indicating the temporal constraints can improve both final position and trajectory probability prediction. Compared with temporal consistency, spatial consistency has less effect on models since we only enforce this constraint in the trajectory refinement stage. Finally, the Teacher-Target Constraints significantly increases performance, manifesting its effectiveness in helping training convergence.

**Temporal Consistency Factors.** We study the factors in the matching problems, including similarity and matching strategies. As shown in Tab. 3, both Hungarian and Bidirectional matching show their advantages over the single direction matching. Although Hungarian matching can ensure the

Table 3: Ablation study on matching factor for temporal consistency. In this experiment, we remove the Teacher-Target Constraints to fairly study the effect

| Matching Strategy | Similarity | K=1 | | | K=6 | | |
|---|---|---|---|---|---|---|---|
| | | minADE | minFDE | MR | minADE | minFDE | MR |
| Forward | ADE | 1.25 | 2.70 | 0.46 | 0.670 | 0.982 | 0.089 |
| | FDE | 1.24 | 2.69 | 0.46 | 0.668 | 0.980 | 0.088 |
| Backward | ADE | 1.25 | 2.70 | 0.46 | 0.670 | 0.982 | 0.089 |
| | FDE | 1.24 | 2.68 | 0.46 | 0.667 | 0.958 | 0.085 |
| Bidirectional | ADE | **1.22** | **2.67** | 0.446 | 0.666 | 0.972 | 0.087 |
| | FDE | **1.22** | **2.67** | **0.445** | **0.653** | **0.954** | **0.084** |
| Hungarian | ADE | 1.24 | 2.69 | 0.46 | 0.668 | 0.975 | 0.088 |
| | FDE | 1.23 | 2.69 | 0.45 | 0.660 | 0.968 | 0.088 |

Table 4: Ablation study results on the teacher target number $J$

| Teacher Target Num | K=1 | | | K=6 | | |
|---|---|---|---|---|---|---|
| $J$ | minADE | minFDE | MR | minADE | minFDE | MR |
| 1 | 1.29 | 2.82 | 0.50 | 0.70 | 1.03 | 0.104 |
| 3 | 1.28 | 2.80 | 0.48 | **0.69** | 1.02 | 0.10 |
| 6 | **1.26** | **2.77** | **0.47** | **0.69** | **1.01** | **0.09** |

Table 5: Ablation study of consistency constraints and Teacher Target Constraints on different state-of-the-art methods on Argoverse validation set. Performance for methods without constraints is obtained from corresponding papers or our reproduction

| Method | Consistency | TTC | K=1 | | K=6 | |
|---|---|---|---|---|---|---|
| | | | minADE | minFDE | minADE | minFDE |
| LaneGCN (Liang et al., 2020) | × | × | 1.35 | 2.97 | 0.71 | 1.08 |
| | ✓ | × | **1.29** | **2.80** | **0.68** | **1.00** |
| | × | ✓ | **1.30** | **2.88** | **0.69** | **1.04** |
| TPCN (Ye et al., 2021) | × | × | 1.34 | 2.95 | 0.73 | 1.15 |
| | ✓ | × | **1.27** | **2.79** | **0.69** | **1.04** |
| | × | ✓ | **1.30** | **2.86** | **0.69** | **1.09** |
| mmTransformer (Liu et al., 2021) | × | × | 1.38 | 3.03 | 0.71 | 1.15 |
| | ✓ | × | **1.31** | **2.83** | **0.68** | **1.02** |
| | × | ✓ | **1.29** | **2.80** | **0.68** | **1.04** |
| DenseTNT (Gu et al., 2021) | × | × | 1.36 | 2.94 | 0.73 | 1.05 |
| | ✓ | × | **1.25** | **2.81** | **0.68** | **0.98** |
| | × | ✓ | **1.30** | **2.82** | **0.69** | **1.00** |

one-to-one matching relationship, it is sensitive to the similarity metric and numerical precision, both of which are not stable in the early training stage. In contrast, bidirectional matching with the FDE similarity metric nearly achieves the best results across all the evaluation metrics. Meanwhile, we also conduct experiments to find the best time-shift value $s$ in the temporal consistency. The details can be found in appendix 6.

**Number of Teacher Targets.** As shown in Tab. 4, more teacher targets could bring better performance. Compared with $J = 1$, 6 teacher targets bring an extra nearly $1\%$ improvements. However, the marginal improvement decreases significantly so we finally choose $J = 6$.

### 4.4 Generalization Capability

To verify the generalization capability of Dual Consistency Constraints and Teacher Targets Constraints, we also apply them to different models with state-of-the-art performance to show that they can be plugin-in training schemes.

**Consistency Component.** As shown in Tab. 5, our dual consistency constraints can effectively improve the performance of models regardless of their representations through the training phase. There is a noticeable improvement of over $5\%$ on every metric, especially for minFDE.

**Teacher Target.** Teacher-Target Constraints is another general training trick that can be widely used in other frameworks. In Tab. 5, we also verify its effectiveness on other public methods. Methods with Teacher-Target Constraints have nearly over $3\%$ improvement in all metrics. For the original DenseTNT (Gu et al., 2021), we replace its original handcrafted optimization for teacher goal targets with our self-ensembling teacher targets. This strategy brings an over $5\%$ increase in performance, demonstrating the better quality of the self-ensembling teacher targets than handcrafted optimizations and estimation.

### 5 Conclusion

In this work, we propose MISC, an effective architecture for the motion forecasting task that explicitly models the multi-modality. We also impose dual consistency regularization on both spatial and temporal domains to leverage the potential of self-supervision, which has been ignored by previous efforts. Besides, we explicitly model the multi-modality by providing supervision with powerful self-ensembling techniques. Experimental results on the Argoverse motion forecasting dataset show the effectiveness of our approach and generalization capability to other methods.

## REPRODUCIBILITY STATEMENT

We use the publicly available Argoverse Dataset (Chang et al., 2019) available at `https://www.argoverse.org/av1.html#forecasting-link`. Dataset preprocessing is shown in 4.1. Training process is in Appendix A.2. And the model architecture is illustrated in the Sec. 3.1 and Appendix A.1.

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

# A APPENDIX

## A.1 MODEL DETAILS

We provide the detailed network architecture of our MISC in Fig. 5. We use TPCN (Ye et al., 2021) as our backbone. The feature extraction consists of 4 spatial modules and 4 dynamic temporal learning layers same as TPCN. Before the prediction header, we calculate the mean features and remove map instances features. For the spatial module, the point representation utilizes PointNet++ (Qi et al., 2017) with neighborhood radius of $[0.2m, 0.4m, 0.8m]$, while the voxel representation uses Sparse BottleNeck. We use all the points in this process without any sampling. More details about backbone can be found in TPCN (Ye et al., 2021).

## A.2 TRAINING DETAILS

We train MISC for $50$ epochs using a batch size of $32$ with Adam (Kingma & Ba, 2014) optimizer with an initial learning rate of $0.001$, which is decayed every $15$ epochs in a ratio of $0.1$.

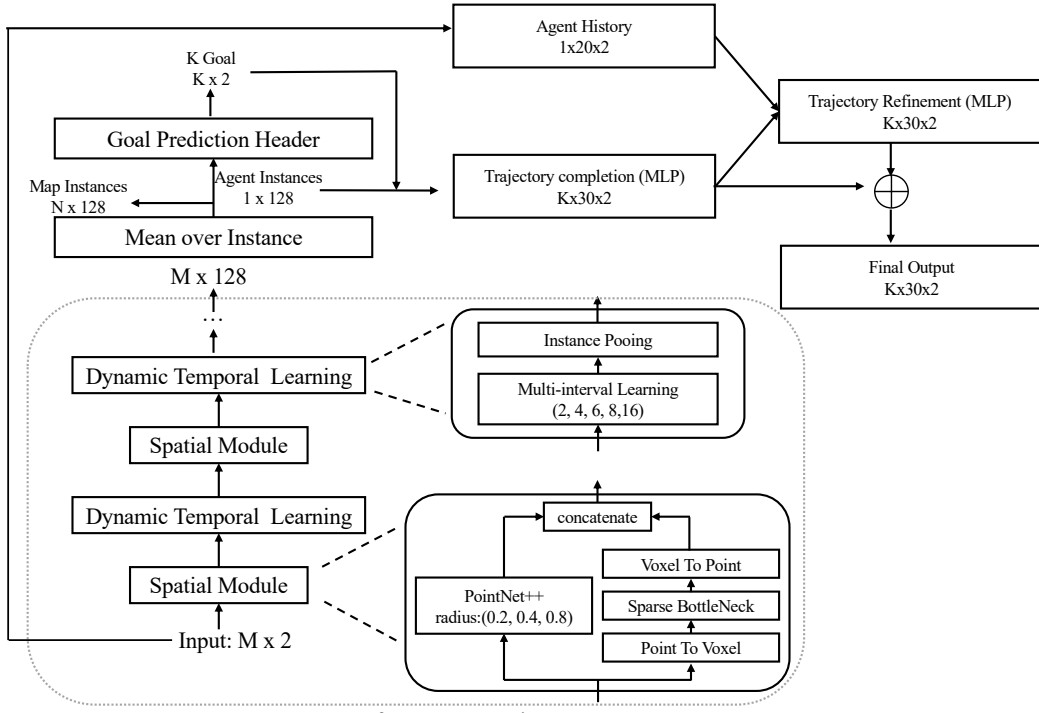

Figure 5: Detailed illustration of our MISC.

## A.3 ABLATION STUDY

### A.3.1 TEMPORAL CONSISTENCY

Meanwhile, we also conduct experiments to find the best time-shift value $s$ in the temporal consistency. As shown in Tab. 6, choosing time shift $s = 1$ has already achieved decent performance, with five out of six metrics ranking the first. Further increasing the $s$ will not bring much performance gain since the driving behavior could change a lot with large $s$.

We use the average L2 distance among all predicted trajectory waypoints to measure the temporal consistency. As shown in Fig. 6, our model without temporal consistency will have large inconsistency even though the time shift $s$ is small, which may lead to unstable behavior for the downstream

Table 6: Ablation study results of time-shift $s$ used by temporal consistency

| Time shift | K=1 | | | K=6 | | |
|---|---|---|---|---|---|---|
| $s$ | minADE | minFDE | MR | minADE | minFDE | MR |
| 1 | **1.22** | **2.67** | **0.444** | **0.653** | **0.954** | 0.084 |
| 2 | 1.23 | **2.67** | **0.444** | 0.654 | 0.958 | **0.082** |
| 3 | 1.25 | 2.69 | 0.445 | 0.662 | 0.964 | 0.085 |
| 4 | 1.25 | 2.70 | 0.446 | 0.667 | 0.969 | 0.086 |

Figure 6: The L2 distance in our model varies with the time shift $s$.

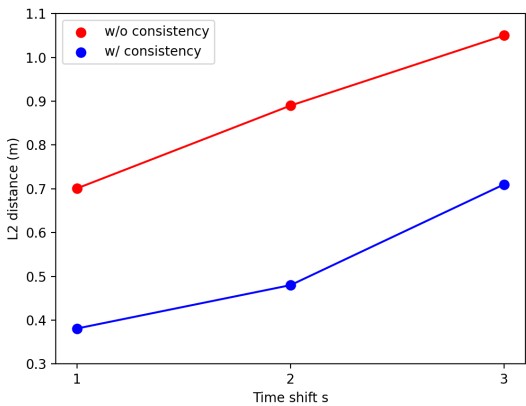

task such as planning. With temporal consistency constraints, there is a significant improvement for the L2 distance divergence, demonstrating the effectiveness of our method.

### A.3.2 SPATIAL CONSISTENCY

Furthermore, we also measure the spatial inconsistency against flipping and Gaussian noise with zero mean and standard deviation of 15cm. The average spatial inconsistency will be 19.3cm, while the number decreases to 10.2cm with our spatial consistency constraint.

### A.3.3 COMPONENT STUDY

We provide a controlled experiment to verify the effectiveness of the proposed method when turning both Dual Consistency Constraints and Teacher-Target Constraints on at the same time shown in Tab. 7. With both modules on, the performance of all the methods benefits a lot, about nearly 7%, demonstrating the generalization capability and effectiveness of our approach. It also shows that these two modules can be independently helpful.

### A.3.4 RESULTS ON WAYMO DATASET

Table 8: Quantitative results on the validation set of the Waymo Open dataset motion prediction task.

| Method | minADE↓ | minFDE↓ | Miss Rate↓ | mAP↑ |
|---|---|---|---|---|
| Baseline (Ettinger et al., 2021) | 0.675 | 1.349 | 0.183 | 0.268 |
| KEMP (Lu et al., 2022) | 0.5691 | 1.1993 | 0.1458 | 0.394 |
| SceneTransformer (Ngiam et al., 2021) | 0.613 | 1.22 | 0.157 | 0.284 |
| Ours | **0.54** | **1.11** | **0.128** | **0.41** |

Table 7: Results of consistency constraints and Teacher-Target Constraints (TTC) supervision on different state-of-the-art methods on Argoverse validation set. Performance for methods without consistency constraints is obtained from corresponding papers or our reproduction.

| Method | Consistency & TTC | K=1 | | K=6 | |
|---|---|---|---|---|---|
| | | minADE | minFDE | minADE | minFDE |
| LaneGCN (Liang et al., 2020) | × | 1.35 | 2.97 | 0.71 | 1.08 |
| | ✓ | **1.25** | **2.71** | **0.66** | **0.98** |
| TPCN (Ye et al., 2021) | × | 1.34 | 2.95 | 0.73 | 1.15 |
| | ✓ | **1.23** | **2.70** | **0.67** | **1.00** |
| mmTransformer (Liu et al., 2021) | × | 1.38 | 3.03 | 0.71 | 1.15 |
| | ✓ | **1.25** | **2.77** | **0.67** | **0.99** |
| DenseTNT (Gu et al., 2021) | × | 1.36 | 2.94 | 0.73 | 1.05 |
| | ✓ | **1.23** | **2.71** | **0.66** | **0.95** |

We provide some quantitative results on the validation set of the Waymo Open dataset motion prediction task (Ettinger et al., 2021), shown in Tab. 8. Compared with KEMP (Lu et al., 2022) and SceneTransformer (Ngiam et al., 2021), we also achieve very promising results and show comparable improvement, demonstrating the effectiveness of our approach.

### A.3.5 Ablation Study on Waymo Dataset

Since the scale and object types in waymo dataset and argoverse dataset are different, we conduct experiments to find the best time shift $s$ for each class on Waymo Dataset. As shown in Tab. 9, best time shift for vehicle and cyclist will be 1, while the value will be 2 for pedestrian class. To achieve the best performance for the overall metrics, we finally choose $s = 1$ in our setting.

Table 9: Ablation study results of time-shift $s$ used by temporal consistency on Waymo Open Motion Dataset motion prediction

| Time shift | minADE↓ | | | minFDE↓ | | | MR↓ | | | mAP↑ | | |
|---|---|---|---|---|---|---|---|---|---|---|---|---|
| | veh | ped | cyc | veh | ped | cyc | veh | ped | cyc | veh | ped | cyc |
| 1 | **0.622** | 0.34 | **0.654** | **1.262** | 0.663 | 1.294 | **0.135** | 0.085 | **0.197** | 0.285 | **0.252** | **0.214** |
| 2 | 0.625 | **0.33** | 0.660 | 1.263 | **0.662** | 1.296 | **0.135** | **0.084** | 0.200 | **0.283** | **0.252** | 0.215 |
| 3 | 0.632 | 0.34 | 0.667 | 1.274 | 0.666 | 1.302 | 0.136 | 0.086 | 0.198 | 0.290 | 0.254 | 0.217 |
| 4 | 0.634 | **0.33** | 0.672 | 1.278 | 0.670 | 1.303 | 0.137 | 0.086 | 0.199 | 0.288 | 0.253 | 0.217 |

### A.4 Results on ETH Dataset

To verify the temporal consistency on the low framerate dataset, we conduct experiments on the ETH Pellegrini et al. (2010) dataset. We report the **ADE** and **FDE** metrics for $t_{pred} = 8$ and $t_{pred} = 12$ respectively. Following the common settings used by previous methods Fang et al. (2020), we use $K = 1$ and $K = 20$. As shown in Tab. 10, our temporal consistency significantly improves the performance. Choosing $s = 1$ works well in most of the evaluation metrics.

### A.5 Model complexity

We provide detailed runtime speed evaluated in a single RTX2080Ti with the model parameters shown in Tab. 11. Compared with other state-of-the-art models, we achieve decent performance without introducing more computation cost.

### A.6 Qualitative Analysis

We provide some visual results of MISC on the the Argoverse (Chang et al., 2019) validation set in Fig. 8 as well as the Argoverse test set in Fig. 9. These qualitative results demonstrate the effectiveness and the high-quality predicted trajectories of our method.

Table 10: Ablation study results of time-shift $s$ used by temporal consistency on ETH Dataset

| Time shift | Dataset | K=1 | | K=20 | |
|---|---|---|---|---|---|
| | | ADE | FDE | ADE | FDE |
| 0 | **ETH** | 0.69 / 0.98 | 1.30 / 1.98 | 0.51 / 0.79 | 1.05 / 1.66 |
| | **HOTEL** | 0.27 / 0.33 | 0.46 / 0.55 | 0.20 / 0.25 | 0.36 / 0.44 |
| 1 | **ETH** | **0.65** / 0.93 | **1.22 / 1.86** | **0.47 / 0.73** | **0.97 / 1.55** |
| | **HOTEL** | **0.23** / 0.29 | **0.42** / 0.50 | **0.18 / 0.23** | **0.33 / 0.42** |
| 2 | **ETH** | **0.65 / 0.92** | 1.23 / 1.88 | 0.48 / **0.73** | 1.00 / 1.56 |
| | **HOTEL** | 0.24 / **0.27** | 0.43 / **0.49** | **0.18** / 0.25 | 0.34 / **0.42** |
| 3 | **ETH** | 0.66 / 0.93 | 1.24 / 1.89 | 0.48 / **0.73** | 0.98 / 1.57 |
| | **HOTEL** | 0.24 / 0.30 | 0.43 / 0.52 | 0.19 / 0.24 | 0.34 / 0.44 |
| 4 | **ETH** | 0.66 / 0.94 | 1.23 / 1.89 | 0.49 / 0.74 | 0.99 / 1.58 |
| | **HOTEL** | 0.25 / 0.31 | 0.44 / 0.51 | 0.20 / 0.25 | **0.33** / 0.44 |

Table 11: The number of parameters and running time.

| Method | Param (M) | Speed (ms) |
|---|---|---|
| LaneGCN | 3.7 | 55 |
| DenseTNT | **1.1** | 40 |
| mmTransformer | 2.6 | **34** |
| Ours | 3.6 | 36 |

## A.7 FAILURE CASES

We also present some failure cases on the validation set in Fig. 7. Some possible reasons are:

- The ground-truth labels contain some noises. Since the ground-truth labels are obtained from tracking, there may be some id switches, leading to the sudden perturbation of the agents' location (e.g., the first and third example in the second row of Fig. 7). Under these scenarios, the predicted trajectories from MISC are more reasonable and stable without large jerks.

- The multi-modality problem. In some situations, MISC can not predict the intention perfectly without enough motion and map information. The first and third example in the first row of Fig. 7 demonstrate this phenomenon. The agent makes a lane change decision without many hints in the historical information. Thus, this can be furthered improved by introducing more map constraints.

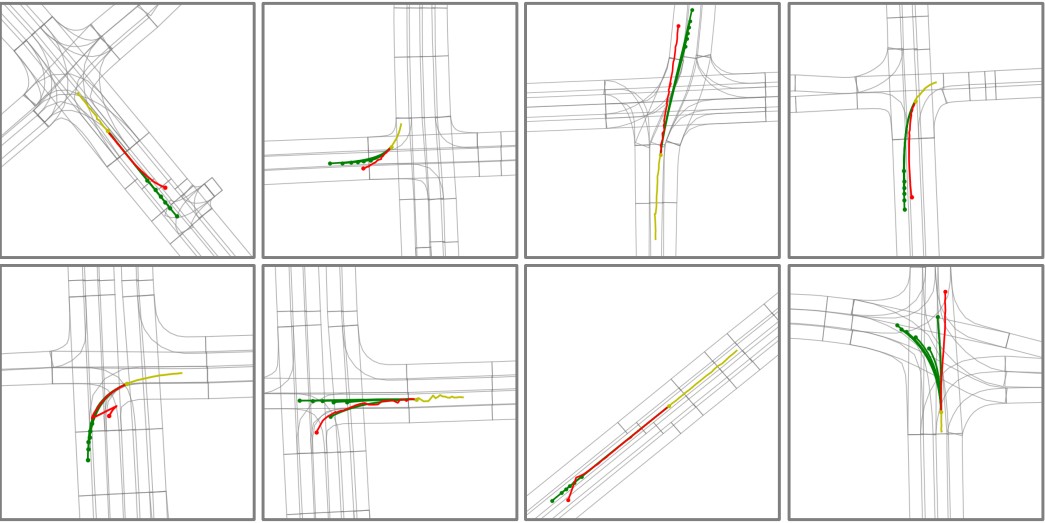

Figure 7: Failure cases on the Argoverse validation set. The target agent's past trajectory is in yellow, predicted trajectory in green, and ground truth in red.

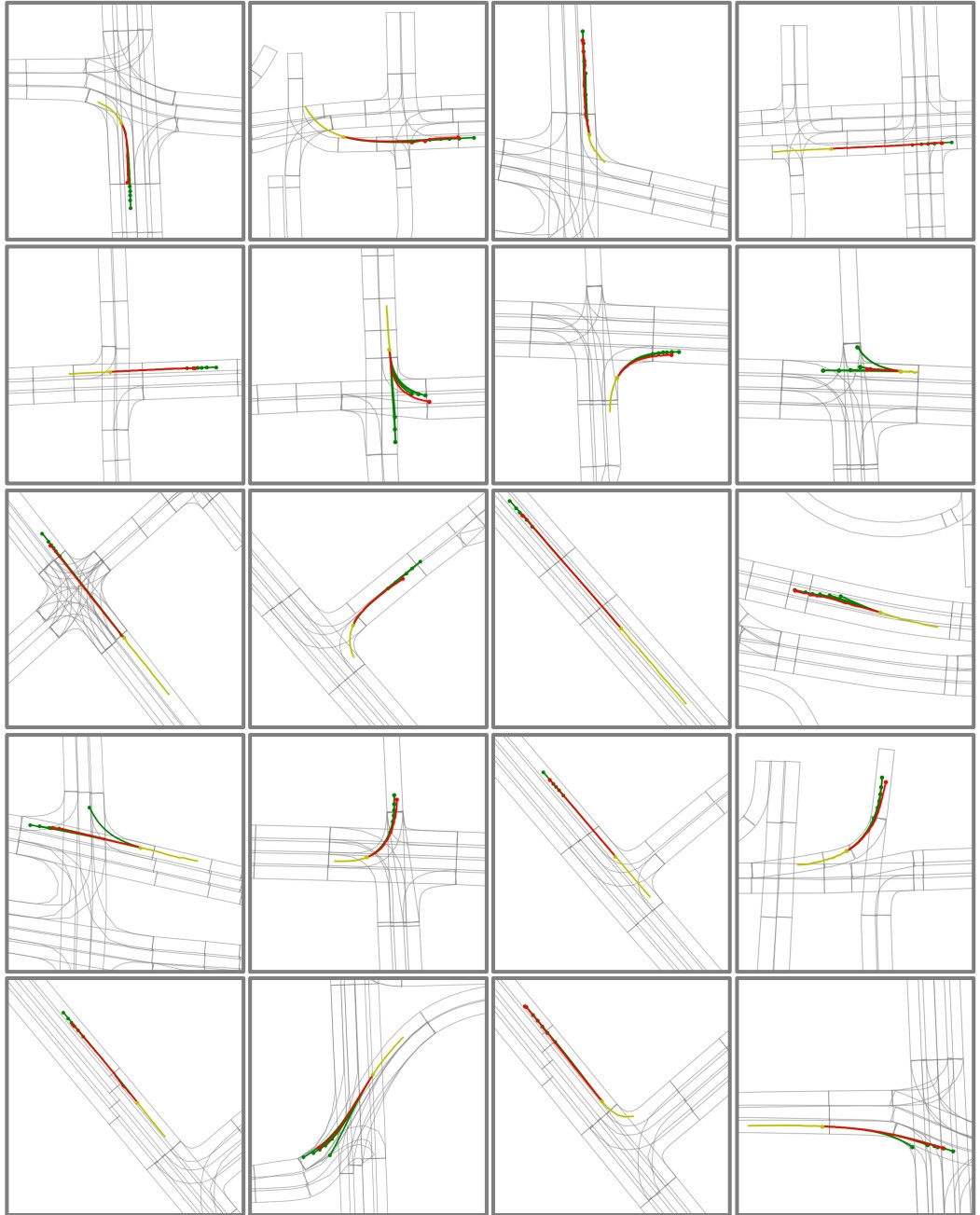

Figure 8: The motion forecasting results on the Argoverse validation set. The target agent's past trajectory is in yellow, predicted trajectory is in green, and ground truth is in red.

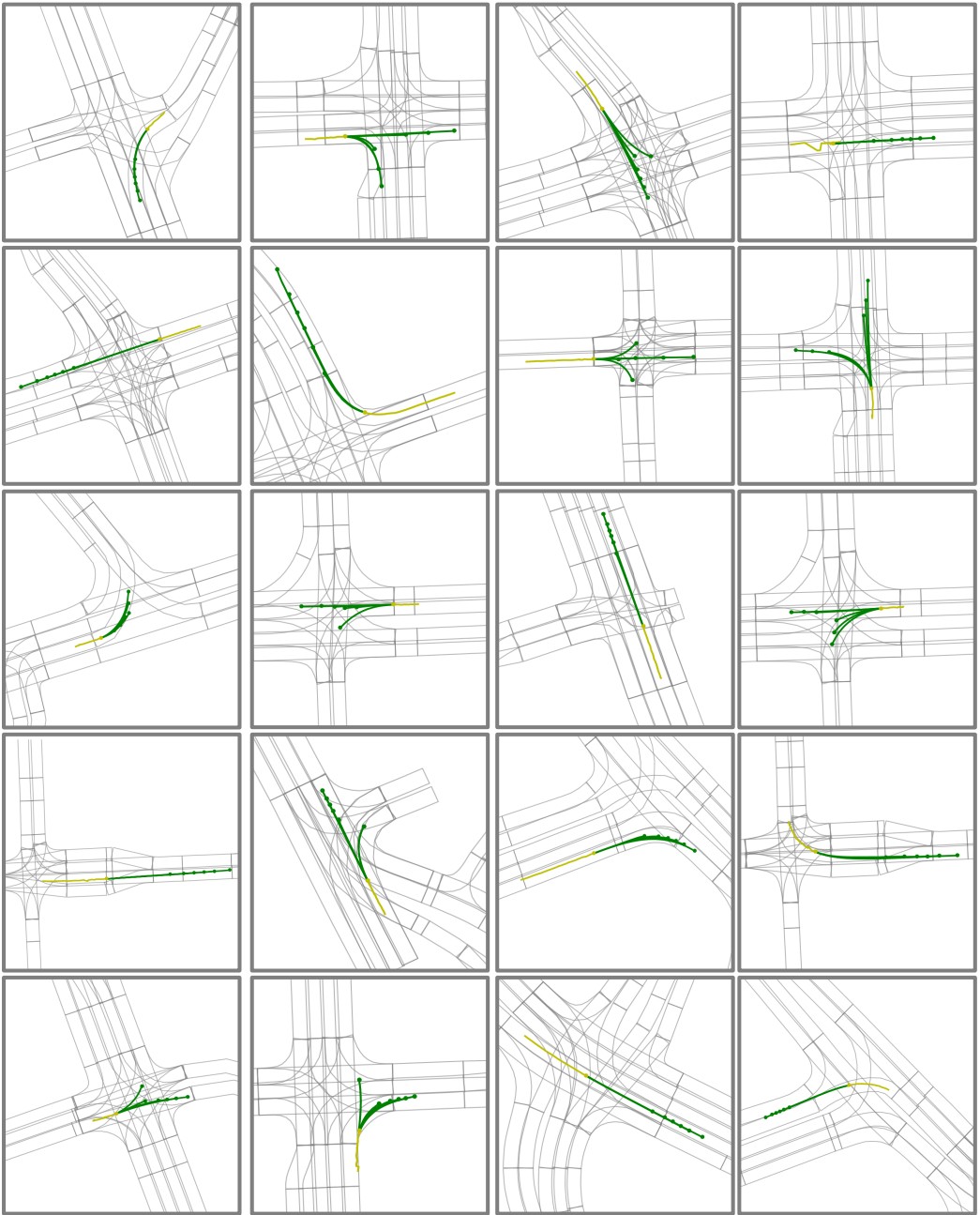

Figure 9: The motion forecasting results on the Argoverse test set. The target agent's past trajectory is in yellow and predicted trajectory in green.

