# OpenReview forum: "Bootstrap Motion Forecasting With Self-Consistent Constraints"
_ICLR.cc/2023/Conference — Submitted to ICLR 2023_

### Official Review · Reviewer_MgTW · 2022-10-14

**Confidence:** 3
**Correctness:** 3
**Technical Novelty And Significance:** 2
**Empirical Novelty And Significance:** 3
**Recommendation:** 8

**Clarity, Quality, Novelty And Reproducibility:**

- **Clarity**: Overall the paper is written clearly, though the significance of contributions is not currently very much clear.
- **Quality**: The quality is ok but not enough to me, due to some limited experimental evaluation.
- **Novelty**: As far as I have checked, the proposed spatiotemporal constraint for trajectory forecasting seems to be a novel idea.
- **Reproducibility**: Most implementation details are presented in the paper, although there is no guarantee that the presented results can be reproduced perfectly due to the absence of code submissions and random seed settings.

**Strength And Weaknesses:**

## Strong points
- Overall, the paper is well written and easy to follow.
- The motivation to introduce the proposed spatiotemporal constraints is valid, and the specific approach to realize that key idea seems straightforward and technically sound. As far as I have checked, introducing such spatiotemporal constraints for trajectory forecasting seems new.
- The proposed method outperforms many recent methods on the Argoverse leaderboard. The ablation study further shows that each of the technical components in the proposed method contributes to the final forecasting performance.

## Weak points
- The paper could have become stronger if the method is evaluated on various datasets other than Argoverse, which involve not only vehicles but also pedestrians, together with more detailed analysis and discussions on how s time shift s should be determined. I think so because the appropriate setting of s can depend on video framerates and the frequency of sudden changes in motion. As s becomes larger or for a lower framerate videos, the temporal constraint could enforce inappropriate matching and may result in performance degradation as implied in Table 6. On the other hand, when s is too small or video framerates are too high, the temporal constraint will do little. It also remains unclear how well the proposed method works when trajectories involve many sudden changes of motion speeds and directions, like those of pedestrians. Currently we can learn from the paper only that s=1 worked the best for Argoverse, which is not very much informative considering the potential applications of the proposed method to other datasets.
- The limitation and failure cases of the proposed method are not discussed, which I believe are crucial to judge the significance of the method. If the proposed method is applicable just for smooth vehicle trajectories recorded at a framerate within a certain narrow range, the proposed work could have a limited applicability and significance. If not, I think that this paper presents a nice contribution with a simple idea that works effectively for various situations.

## Additional feedback
- I would expect qualitative results to visualize not only the proposed method but also other baselines.

**Summary Of The Paper:**

This paper presents a new method for trajectory forecasting. The proposed method consists of two ideas. (1) It introduces a new constraint such that (1a) predicted trajectories from consecutive inputs should not be different (temporal consistency) and (1b) the predicted trajectories should also be consistent regarding spatial transformations due to flipping and random noise (spatial consistency). (2) It further introduces a teacher-target constraint by enforcing multiple predictions from different training procedures to be consistent. Experimental results on the Argoverse dataset demonstrates the effectiveness of the proposed method over state-of-the-art methods.

**Summary Of The Review:**

Currently I'm very neutral about this paper. It proposes a simple learning strategy involving several ideas, which performs really nice on the Argoverse leaderboard. However, it is not clear how applicable the proposed method is to a wide variety of situations, such as lower/higher framerate videos, or videos with other moving agents such as pedestrians. Limitations and failure cases are not discussed in detail, leaving it difficult to judge the significance of the proposed approach with the current manuscript.

---
Post-discussion comment: As my initial concerns above were resolved by the active discussion and paper revision with the authors, I would like to upgrade my score. Nevertheless,

---

> ### Author Response · Authors · 2022-11-14
> **To Reviewer 4**
>
> **Q1:** *The paper could have become stronger if the method is evaluated on various datasets other than Argoverse*
>
> **A1:** We have included the experiments on the Waymo open dataset in Appendix **3.4**. Compared with other SOTA methods, our MISC has also shown superior performance.
>
> **Q2:** *Settings on the time sift factor.*
>
> **A2:** Thanks for your appreciation for our temporal consistency. We have conducted an ablation study for time shift $s$ in Appendix **3.1**. Due to the page limits, we have included some other ablation studies about the choices of some key hyper-parameters in the appendix.
>
> **Q2:** *Failure cases and discussion.*
>
> **A2:** Thanks for pointing out that. We have included some failure cases with some possible reasons in the Appendix. **6**.
>
> **Q3:** *The limitation of the proposed method*
>
> **A3:** Although the proposed method achieves state-of-the-art performance and can be a general component for motion forecasting tasks, it still has several drawbacks: 1). Our method lacks strong mathematical proof to support the proposed components; 2). It requires more training resources, including multiple GPUs for generating teacher targets by self-ensembling. Meanwhile, when training is equipped with temporal consistency, the training time and memory usage will nearly double.

---

> > ### Comment · Reviewer_MgTW · 2022-11-14
> > **Response**
> >
> > Thank you very much for the response and revision! While I understood failure cases and limitations, I am not yet perfectly convinced by A1 and A2 and my original concern remains unresolved: *it is not clear how applicable the proposed method is to a wide variety of situations, such as lower/higher framerate videos, or videos with other moving agents such as pedestrians.*
> >
> > If I understood it correctly, both Argoverse and Waymo datasets are recorded 10fps and only vehicle trajectories are predicted. It is not obvious if “s=1” is the best parameter for other datasets with different fps or with different moving targets such as pedestrians. For example, if we need to tune the s parameter for each framerate and for each type of moving target independently, that will reduce the usefulness of the proposed approach.
> >
> > To resolve this concern, would it be possible to perform additional experiments on human trajectory dataset such as UCY/ETH or Stanford drone dataset?

---

> > > ### Author Response · Authors · 2022-11-14
> > > **Response to Reviewer 4**
> > >
> > > Thanks for your feedback. The motion prediction task for Waymo datasets contains three types of objects to be predicted, including pedestrian, vehicle and cyclist. Our results shown in Appendix **3.4** are the average results of these three types according to the official evaluation toolkit.
> > >
> > > As for the additional experiments on human trajectory dataset such as UCY/ETH or Stanford drone dataset, we will try to work on one of them but that might cost us some time and efforts.

---

> > > > ### Comment · Reviewer_MgTW · 2022-11-14
> > > > **Response**
> > > >
> > > > Thanks for the quick reply. If the Waymo dataset already contains moving targets other than vehicles, it’s great and would it be possible to show how performances change depending on the types of moving targets (rather than just showing their average) for each candidate of the s parameter? That would also answer my second question: appropriate s against types of moving targets.
> > > >
> > > > On the extra experiments, I wound not expect results for all of the datasets suggested above, as what I essentially wanted to know is how the selection of s is robust against different framerates.
> > > >
> > > > Thanks!

---

> > > > > ### Author Response · Authors · 2022-11-15
> > > > > **Response to Reviewer 4**
> > > > >
> > > > > We include the detailed ablation study for time shift value $s$ on waymo dataset for each class, shown in Appendix **3.5** in our latest revision. The best value for each class may be a little different. One possible reason is that the dynamics for each category is different. Hope it is clear to you.

---

> > > > > ### Author Response · Authors · 2022-11-17
> > > > > **Response to Reviewer 4**
> > > > >
> > > > > We have updated the revision version. And experiments on ETH dataset have been included in Tab.**9** in the Appendix.**4**. In summary, the *s* finally chosen is similar to the value in other dataset. Hope it is clear to you. Due to the time limit,  we will revise some of the experiments parts to make the table and contents more clear in the final version.

---

> > > > > > ### Comment · Reviewer_MgTW · 2022-11-17
> > > > > > **Thank you**
> > > > > >
> > > > > > I thank the authors for their active discussion and paper revision. As the ETH result was added, I think that the proposed idea was confirmed to be effective in a variety of situations (both vehicles and pedestrians and several different framerates). My remaining concern is about the paper organization; please consider (if the paper is accepted) to summarize all these results in the main body of the paper rather than in its appendix, so that readers can understand that the paper is presenting a simple solution that works well in various situations. I think it's possible given some space left in the current manuscript.
> > > > > >
> > > > > > Anyway my initial concerns had been addressed and I would like to raise my score.

---

### Official Review · Reviewer_xHGP · 2022-10-21

**Confidence:** 4
**Correctness:** 3
**Technical Novelty And Significance:** 2
**Empirical Novelty And Significance:** 3
**Recommendation:** 5

**Clarity, Quality, Novelty And Reproducibility:**

I think the paper is overall clear but missing important analysis, which undermines the contribution of the work significantly. Here are some questions I would like to ask.

Questions:
1. What is the criterion when choosing spatial permutation function Z?  It seems such a perturbation of the input shall not lead to a significant input change.
2. How does spatial consistency distinguish itself from data augmentation?
3. Why do we need a dedicated refinement module for spatial consistency?
4. Any evidence to show that the learned prediction would lead to more stable behaviors when coupled with down streams tasks like planning?
5. Would the time consistency impose significantly more computation costs during training? To calculate time consistency, it seems the same trajectory would be trained multiple times.
6. How should the teacher model be trained to ensure enough diversity?

Minor points:
1. By reading the introduction, it is not entirely clear to me what consistency refers to. It can be more explicit if a concrete example is included to illustrate the concept.
2. It is difficult for me to understand what is the notion of spatial consistency, in particular, what is the interpretation of consistency here.
3. Is the bidirectional matching strategy the same as chamfer distance?

**Strength And Weaknesses:**


Strength:

1. The proposed method would only increase the training cost by a reasonable amount of training resources and does not make deployment more computationally expensive.

2. The authors do a great job of demonstrating the framework's efficacy through an extensive comparison with the current state of the art.

3. The framework is tested with existing approaches, making the result even more convincing.

Weaknesses:

1. I am not sure whether it is a good idea to put different modules under the same umbrella, namely, self-consistency since the connection between components presented in the paper is loose. Each of them seems ad-hoc, thus, providing limited insights into the overall framework.

2. The paper emphasizes too much on the result comparison, yet, misses important analysis. While there is a ‘standard’ component analysis tailored to understand ‘how much' each component contributes to the final improvement, not much analysis is given to ‘how’ each component helps. For example, in section 3.2, the paper states ‘Compared with data augmentation, it is the explicit regularization”. It is easy for the reader to identify the difference here, but much harder for the reader to understand which one would be better and why it is better. I think the experiment part suffers from the issue most. The experiment results do not reveal much information about the design choice of the framework.



**Summary Of The Paper:**

This paper presents Motion forecastIng with Self-consistent Constraints (MISC), a framework that allows bootstrapping the performance of motion prediction without additional data. There are three different components proposed in the paper, namely, temporal consistency, which aims to enforce the prediction model to output consistent results given two overlapping observations. Spatial consistency aims to enforce the refinement module to generate the same results given perturbation on the input. The teacher-target constraints allow us to distill a model from multiple teacher models with different initializations and the same architecture.



**Summary Of The Review:**

Given the limited insight provided in the paper, I think the technical contribution of the work is not significant enough. I lean towards rejection for now.

---

> ### Author Response · Authors · 2022-11-14
> **To Reviewer 3**
>
> **Q1:** *The connection between different components presented in the paper is loose under the same umbrella, namely, self-consistency.*
>
> **A1:** The overall motivation of our MISC is to explore the self-consistent constraints for the motion forecasting task to bootstrap the performance without extra labels. There are two ways that can achieve this goal. One is the spatial-temporal consistency due to the streaming and physical properties of this task, namely the inner model consistent constraint. The other way is the intra-model consistency constraint, which aims to introduce the regularization from more powerful models through a self-ensembling way. Meanwhile, the multi-teacher-target can also alleviate the multi-modality training problem. In summary, our MISC consists of two self-consistent constraints from the inner-model and intra-model perspectives. To our best knowledge, we are the first work that combines these two kinds of properties with a motion forecasting task to explore the streaming and multi-modality property of this task as self-consistent constraints. Moreover, our proposed modules can be simple and effective to be used as a plug-in component. We will modify the paper writing accordingly and clarify our motivation for MISC in the revised version.
>
> **Q2:** *The paper emphasizes too much on the result comparison, yet, misses important analysis. While there is a ‘standard’ component analysis tailored to understand ‘how much' each component contributes to the final improvement, not much analysis is given to ‘how’ each component helps.*
>
> **A2:** To verify the effectiveness of each proposed module and how each component contributes to the final improvement, we have conducted ablation studies in Tab.**2**. We show the improvements brought by each proposed module and the baseline performance. We also provide some analysis in Component Study in Sec.**4.3**.
>
> **Q3:** *How does spatial consistency distinguish itself from data augmentation*
>
> **A3:** Data augmentation  does not enforce any losses for the predictions from two augmented inputs. In contrast, we explicitly enforce the regularization  motivated by contrastive learning.
>
> **Q4:** *Why do we need a dedicated refinement module for spatial consistency*
>
> **A4:** As shown in Tab.**2**, trajectory refinement module will help refine the trajectory. And currently, most of the prediction methods, including DenseTNT are built upon in a two-stage manner.
>
> **Q5:** *Any evidence to show that the learned prediction would lead to more stable behaviors when coupled with down streams tasks like planning?*
>
> **A5:** Thanks for pointing out that. We also include experiments to demonstrate the effectiveness of temporal consistency to ensure stability. In Appendix.**3.1**, we make streaming predictions under time-shift $s$ and measure the difference between the predictions. We can observe that models without temporal consistency will have large inconsistent predictions even though the time shift $s$ is small. However, our temporal consistency significantly minimizes this difference.
>
> **Q6:** *Would the time consistency impose significantly more computation costs during training?*
>
> **A6:** Yes, that is also a limitation of our work. But our method is free-cost for inference, which helps improve the performance without computational cost when deployed.
>
> **Q7:** *How should the teacher model be trained to ensure enough diversity.*
>
> **A7:** As we discussed in the **Sec. 3.3**, we just use the WTA strategy to train the initial models and use the KMeans method to obtain the teacher model. As for diversity, we use a similar idea in DenseTNT[1] and GOHOME[2]. We calculate the L2 distance between the predicted trajectories and then count the number of distances beyond a threshold ($2$m). A larger number means larger diversity.
>
> **Q8:** *It is difficult for me to understand what is the notion of spatial consistency, in particular, what is the interpretation of consistency here*
>
> **A8:** As shown in Fig. **2**, the consistency in general means that under some disturbance, our network should be stable enough. Spatial consistency is motivated by contrastive learning. The paired features from one input with different augmentations should share consensus to be similar enough.
>
> **Q9:** *Is the bidirectional matching strategy the same as chamfer distance.*
>
> **A9:** Actually, they are not the same. Bidirectional matching is just a cyclic way for matching, considering forward and backward matching.

---

> > ### Author Response · Authors · 2022-11-14
> > **citations**
> >
> > [1] Junru Gu, Chen Sun, and Hang Zhao. Densetnt: End-to-end trajectory prediction from dense goal
> > sets. In Proceedings of the IEEE/CVF International Conference on Computer Vision, pp. 15303–
> > 15312, 2021
> >
> > [2] Thomas Gilles, Stefano Sabatini, Dzmitry Tsishkou, Bogdan Stanciulescu, and Fabien Moutarde.
> > Gohome: Graph-oriented heatmap output for future motion estimation. In 2022 International
> > Conference on Robotics and Automation (ICRA), pp. 9107–9114. IEEE, 2022

---

> > ### Comment · Reviewer_xHGP · 2022-12-01
> > **Response**
> >
> > I thank the authors for their response and I appreciate all the effort that the authors have put into the rebuttal (e.g. plenty of additional experiments)!
> >
> > The rebuttal addressed some of my concerns. Yet, I am still struggling with the main motivation of the overall framework, which remains a bit 'artificial' to me. I agree with Reviewer mCiK that the main novelty seems to be the temporal consistency constraint. Thus, I would suggest the author focus more on it and go deeper. For example, what is the underlying issue that causes inconsistency, and why the proposed temporal consistency constraint is a good option to tackle that from first principles?
> >
> > I have raised my score. Unfortunately, I am unable to recommend the paper for acceptance in its current form.

---

### Official Review · Reviewer_mCiK · 2022-10-25

**Confidence:** 4
**Correctness:** 3
**Technical Novelty And Significance:** 2
**Empirical Novelty And Significance:** 3
**Recommendation:** 5

**Clarity, Quality, Novelty And Reproducibility:**

The paper is very clear in general and details provided are sufficient to make it reproducible. For novelty, see "summary" section.

One area that is not clear is: how exactly ensembling was done at test time. Experimental section says "We finally use 10 models for
ensembling due to computation resource limits". I assume these 10 models were used, along with K-means to provide 6 targets to train a student model? On the Argo and Waymo leaderboards, most results were obtained using an ensemble of models already. Did you use a single model for the leaderboards? More details here would be helpful.

Eq3 as written is not particularly clear. At this point, Z operation is not well defined -- what is Z^-1 for random noise? And what is "Reg"?

Minor language: "furtherly".

**Strength And Weaknesses:**

Model strengths:
- The temporal consistency constraint is simple, effective and seems novel.
- Distillation (called Teacher - Target Constraints) in the paper), even if it does not appear novel per se, is shown to lead to SOTA (state of the art) results. The ICRA 22 paper by Su et al does not come close experimentally to push a model to SOTA.

Experiment strengths:
- SOTA or close to SOTA results on both Argoverse and Waymo Open Motion Dataset.
- Relevant ablation results on the effect of introduced losses, also showing the constraints benefit models other than TPCN.

Weaknesses:
- Main novelty seems to be the temporal consistency constraint. Given how straightforward it is (predictions from two adjacent steps should be similar), unclear if this is sufficient for acceptance.
- Some relevant work or context is not cited, especially for teacher-target constraints.

**Summary Of The Paper:**

The paper introduces several consistency losses that help improve the performance of behavior prediction models.
* The main innovation is the temporal consistency constraint, which enforces that predictions from adjacent time steps are similar. This is simple, but effective and -- to the best of my knowledge -- novel.
* The spatial constraint is in effect a data augmentation technique -- enforcing robustness to small Gaussian noise to the input trajectories. I am not aware of this being commonly used, however, this is fairly straightforward, and the effect is pretty minor, compared to the temporal consistency constraint. As related work, ChauffeurNet (Bansal et al, 2019) also apply spatial perturbations, when they train a somewhat different type of model.
* A model distillation technique is applied as an additional loss in the model, to improve the results. Model distillation for behavior prediction models is not novel per se. I am aware of at least one instance that is not cited:  "Narrowing the Coordinate-frame Gap in Behavior Prediction Models: Distillation for Efficient and Accurate Scene-centric Motion Forecasting" by DiJia Su and Bertrand Douillard and Rami Al-Rfou and Cheolho Park and Benjamin Sapp, ICRA 2022.

The losses above are applying to a TPCN architecture, augmented with a three-stage predictor head (goals -> trajectories -> trajectory refinement). The above is a fairly straightforward combination of existing architecture (TPCN) and model head ideas.

**Summary Of The Review:**

Strong experimental results. Good ablations. Several other ideas with marginal novelty like distillation. The main novelty -- temporal consistency loss -- is fairly straightforward but effective. Main concern is limited overall novelty in this paper, which seems low for ICLR.

I am on the borderline overall, as I appreciate simple techniques that work. My final rating depends on clarifying details around model ensembles: whether leaderboard submissions were one model, or many. If there were many models, how were those obtained -- were they all trained using the same teacher ensemble? More details and experimental results there would help. Also, doing ablation of the main idea -- temporal and spatial consistency loss -- on the Waymo Open Dataset, which has less input noise, would help as well.

---

> ### Author Response · Authors · 2022-11-14
> **To Reviewer 2**
>
> **Q1:** *how exactly ensembling was done at test time.*
>
> **A1:** In the testing time, we do not use the test time ensemble since we have achieved decent performance when submitting to the Argoverse testing server. We will clarify these details in the revised version.
>
> **Q2:** *Some relevant work or context is not cited, especially for teacher-target constraints.*
>
> **A2:** Thanks for mentioning this related work, and we will cite and discuss it. The preprint of this work with teacher-target constraints appeared on arXiv in April 2022, while the mentioned work was published in May. The goals of our MISC and the mentioned work are similar in using distillation or self-consistent constraints as intra-model consistency for motion forecasting. However, we achieve this goal in different ways. Our MISC utilizes the multi-modality properties through self-ensembling techniques to provide explicit multi-modality supervision. Previous methods only consider one single ground truth, suffering from instability or confusion when dealing with complex road topology. While the mentioned work is based on scene-centric and agent-centric differences. Our MISC and the mentioned work also generate the teacher target in a different way. Our teacher target is from a self-ensembling way, and the mentioned work is from an agent-centric model.
>
> **Q3:** *Minor issues.*
>
> **A3:** We are sorry for the typos and confusion. Function $Z$ is a spatial permutation function like flipping, and Function $Z^{-1}$ is the reverse function like inverse flipping to the original coordinate.
>
> Besides that, we demonstrate our motivations in the answer to [All Reviewers](https://openreview.net/forum?id=7KSeWGIOYM&noteId=xz-LXvevE-W).

---

> > ### Comment · Reviewer_mCiK · 2022-12-02
> > **Response to Authors**
> >
> > > In the testing time, we do not use the test time ensemble since we have achieved decent performance when submitting to the Argoverse testing server.
> >
> > I will assume your Argo result is with a single model then? Please confirm? That would be an unusually strong result for a single model (distillation from a good ensemble could in theory retain its performance...).
> >
> > > The preprint of this work with teacher-target constraints appeared on arXiv in April 2022, while the mentioned work was published in May.
> >
> > I do not think this is a sufficient reason to not cite the work given you are submitting the paper late in 2022.  "Previous methods only consider one single ground truth, suffering from instability or confusion when dealing with complex road topology" --> What do you mean exactly? The Su et al work can distill using a distribution of trajectories.

---

> > > ### Author Response · Authors · 2022-12-05
> > > **Response to Reviewer mCiK**
> > >
> > > **Q1:** test-time model ensemble
> > >
> > > **A1:** We are sorry about our inaccurate and confusing description. We just misunderstand your question. We just want to emphasize that we don't apply different testing-time ensembling methods than what we applied in obtaining the teacher targets. When submitting to the testing server, we ensemble $6$ models trained from the same teacher targets with different random seed and learning rate choices. The following table show one single model results which can also be found in the argoverse leaderboard:
> > >
> > > |brier-minFDE(k=6)|minFDE(k=6)| minADE(k=6)|minFDE|minADE|
> > > | ---|   ---|---|---|---|
> > > |1.7963|	1.1675	|0.7797	|	3.3297|	1.5099
> > >
> > > The results are also very promising especially for a single model results.
> > >
> > > **Q2:** about the mentioned work
> > >
> > > **A2:** We are sorry for our inaccurate and informal description which may cause some misunderstanding. First, we will cite and discuss this good work in the revised version. Second, the timeline we mentioned in the response is to prove that we do not plagiarize the ideas. And the next part we just want to demonstrate the intuitions of our ideas.
> > > The difference between the mention work and our MISC is in the last part. While the mentioned work is based on scene-centric and agent-centric differences. Our MISC and the mentioned work also generate the teacher target in a different way. Our teacher target is from a self-ensembling way, and the mentioned work is from an agent-centric model.

---

> > > > ### Author Response · Authors · 2022-12-05
> > > > **Response to Reviewer mCiK**
> > > >
> > > > **Q3:** about ablation study on the waymo open dataset.
> > > >
> > > > **A3:** We have included some ablation studies on the waymo open dataset for some parameter choices during the rebuttal period. We also conduct some extra experiments on the ETH dataset. Thanks for your valuable comments and we are expecting further discussions with you.

---

> > > > > ### Comment · Reviewer_mCiK · 2022-12-05
> > > > > **Thank you for the clarifications**
> > > > >
> > > > > They help, as your initial response still was unclear on these points.

---

> > > > > > ### Author Response · Authors · 2022-12-08
> > > > > > **Thanks for your response**
> > > > > >
> > > > > > **Q1:** about the experiments and paper structure
> > > > > >
> > > > > > **A1:** As we discussed with **Reviewer MgTW**, we will move some important ablation studies on waymo open dataset into the main paper part from appendix in the final version to clearly demonstrate our methods' generalization capability.

---

### Official Review · Reviewer_5FaF · 2022-11-04

**Confidence:** 5
**Correctness:** 3
**Technical Novelty And Significance:** 3
**Empirical Novelty And Significance:** 3
**Recommendation:** 6

**Clarity, Quality, Novelty And Reproducibility:**

The paper clearly describes the proposed approach. The proposed approach is novel and well-motivated. The authors have provided extensive details for reproducibility which is appreciated.

**Strength And Weaknesses:**

**Strengths**
1. The temporal and spatial consistency constraints are well-motivated, novel and clearly described.
2. Strong performance on the Argoverse dataset show the effectiveness of the proposed approach.
3. Extensive ablation studies are conducted to validate the design choices taken in the paper.
4. The paper is well-written and technically correct.

**Weaknesses**
1. **Single Dataset:** The evaluations are conducted on a single dataset which makes it difficult to judge whether the proposed approach would work in general.
2.  **No experiments for online prediction:** One of the motivations of this work is that temporal consistency could be useful in the real world where the predictions are being made online or in a "stream" as stated in the paper. However, there are no explicit experiments where the paper is evaluated under this setting.
3. **Lack of clear emphasis on main contribution:** The paper introduces several ideas including dual consistency, trajectory refinement and teacher-target constraints. While, I understand the motivation of the individual components and they do help in achieving strong performance on Argoverse, having these individual components makes it difficult to appreciate the main motivation of the work i.e. the dual consistency constraint. Further, it's not clear how the consistency constraint helps for the Argoverse dataset in particular and whether the proposed approach would generalize to other motion forecasting datasets/tasks.

**Summary Of The Paper:**

The paper presents an approach for motion forecasting. Specifically, the paper proposes Motion forecasting with self-consistent constraints (MISC). It introduces temporal and spatial consistency priors to regularize predictions generated by the learning model. Furthermore, the paper also introduces a self-ensembling technique to model multi-modality. The proposed approach is evaluated on the Argoverse dataset.

**Summary Of The Review:**

The work presents an interesting formulation of temporal consistency and applies it to the motion forecasting problem on the Argoverse dataset. The paper achieves strong performance and shows extensive ablation studies to show the effectiveness of each of the proposed components. However, my main concern with this work is that the evaluations are conducted on a single dataset which puts a question mark on the generalization capabilities of the proposed work. Therefore, my initial rating is **reject**. However, I am on the fence and would happily reconsider my recommendation if the authors are able to address the limitations discussed above. In particular, performing evaluations on either more datasets or other motion forecasting tasks could raise the impact of this work significantly.

Update: I appreciate the results on the Waymo Open dataset. I have raised by score to 6. I think the temporal consistency idea is novel and seems to work well. However, I still believe the paper lacks a clear emphasis on a single contribution and seems to combine several independent ideas to improve performance on a single task which reduces the general applicability of the work.

---

> ### Author Response · Authors · 2022-11-14
> **To Reviewer 1**
>
> **Q1:**  *Single Dataset*
>
> **A1:** We are sorry that we include some important experiments and analyses in the appendix due to the limited pages of the paper, which may lead to some confusion and unclear writing. We have included the experiments on the Waymo open dataset in Appendix **3.4**. Compared with other SOTA methods, our MISC has also shown superior performance. Moreover, we also provide some ablation studies on on the Waymo open dataset in Appendix **3.5** .
>
> **Q2:** *One of the motivations of this work is that temporal consistency could be useful in the real world where the predictions are being made online or in a "stream" as stated in the paper. However, there are no explicit experiments where the paper is evaluated under this setting.*
>
> **A2:** Thanks for pointing out that. We also include experiments to demonstrate the effectiveness of temporal consistency. In Appendix **3.1**, we make streaming predictions under time-shift $s$ and measure the difference. Models without temporal consistency will have large inconsistent predictions even though the time shift $s$ is small. Our temporal consistency significantly minimizes this difference in the
>  **Fig.6**. Moreover, we also measure this metric for the spatial consistency in Appendix **3.2**.
>
> **Q3:** *Lack of clear emphasis on main contribution*
>
> **A3:** Our main contribution is the idea of self-consistent constraints. To achieve self-consistent constraints, we focus on two ways. One is the inner-model way which performs spatial-temporal consistency, the other is the intra-model way which regularizes by a self-ensembling method. Both intra and inner models ways do not introduce any extra labels, working in synergy to
> boost the performance of our model. Furthermore, they can be a general and effective training strategy for other motion prediction models.

---

> > ### Comment · Reviewer_5FaF · 2022-12-02
> > **Response to Authors**
> >
> > I thank the authors for their detailed clarifications on my comments.
> >
> > The results on the Waymo open dataset are appreciated. However, I believe they should be moved to the main paper. My concern about single dataset evaluation is addressed by this experiment. However, the experiments in appendix 3.1, don't really reflect a setting where the predictions are made "online" or on a stream of data. Further, I still do not completely understand the connection between different individual components proposed in the paper.  I am raising my score but still cannot confidently recommend this paper in its current form.

---

> > > ### Author Response · Authors · 2022-12-04
> > > **Response to Reviewer 5FaF**
> > >
> > > Thanks for your response. Due to the page limitation of ICLR, we put some experiments in the appendix. And we will re-organize these parts in the final version.
> > >
> > > **Q1:** online or on a stream of data testing.
> > >
> > > **A1:** Thanks for your detailed comments. We understand that the reviewer wants to see more results in a large-scale open-loop or closed-loop testing. To some extents, Appendix 3.1 has shown experimental results for streaming data testing that can be viewed as open-loop testing. In specific, as time-shift $s$ increases, the historical observations slide forward for $s$ steps and make successive predictions to the next $s$ steps, which is a streaming process. As shown in the **Fig.6**, we measure the difference between the overlapped predictions, and  our temporal consistency significantly alleviates this inconsistency issue. However, we can not evaluate the results of the extra $s$ predictions since the current datasets like argoverse only provide groundtruth within a fixed small time window. Moreover, unfortunately, it is hard to obtain a large enough sequence data like a long video clip for the current argoverse and waymo dataset since they only provide a very short data clip for each training and validation data. Recently, We find a latest released nuplan dataset that can be potentially used for both open-loop and closed-loop testing. We will conduct some experiments in these kinds of datasets to further verify the effectiveness of our approach.
> > >
> > > **Q2:** connection between different individual components.
> > >
> > > **A2:** Both two kind of consistencies are based on the properties of motion forecasting task. The overall motivation of our MISC is to explore the self-consistent constraints for the motion forecasting task to bootstrap the performance without extra labels. There are two ways that can achieve this goal. One is the spatial-temporal consistency due to the streaming and physical properties of this task, namely the inner model consistent constraint. The other way is the intra models' consistent constraint, which aims to introduce the regularization from more powerful models through a self-ensembling way. Meanwhile, the multi-teacher-target can also alleviate the multi-modality training problem. In summary, our MISC consists of two self-consistent constraints from the inner-model and intra-model perspectives.

---

### Author Response · Authors · 2022-11-14
**To All Reviewers**

We would like to express our gratitude for the reviewers’ constructive comments on our work. We appreciate that all the reviewers recognize the top performance of our approach on a challenging benchmark and the novelty of our temporal consistency.

The overall motivation of our MISC is to explore the self-consistent constraints for the motion forecasting task to bootstrap the performance without extra labels. There are two ways that can achieve this goal. One is the spatial-temporal consistency due to the streaming and physical properties of this task, namely the inner model consistent constraint. The other way is the intra models' consistent constraint, which aims to introduce the regularization from more powerful models through a self-ensembling way. Meanwhile, the multi-teacher-target can also alleviate the multi-modality training problem. In summary, our MISC consists of two self-consistent constraints from the inner-model and intra-model perspectives. To our best knowledge, we are the first work that combines these two kinds of properties with a motion forecasting task to explore the streaming and multi-modality property of this task as self-consistent constraints. Moreover, our proposed modules can be simple and effective to be used as a plug-in component. We will modify the paper writing accordingly and clarify our motivation for MISC in the revised version.

---

### Decision · Program_Chairs · 2023-01-20

**Decision:**

Reject

**Justification For Why Not Higher Score:**

N/A

**Justification For Why Not Lower Score:**

N/A

**Metareview: Summary, Strengths And Weaknesses:**

The paper proposes to perform motion forecasting with a new technique called MISC, utilizing the temporal and spatial consistency priors. The reviewers generally find the paper has a good performance, and is technically correct. However, most reviewers still have concerns about the quality of the paper draft, and the paper is lacking in-depth analysis. While Reviewer MgTW gave a positive rating, he/she explicitly clarifies rejection is also reasonable. The AC encourages the authors to re-submit the paper to future conferences after clarifying the arguments regarding teacher-target constraint and online prediction.